# Condition, Reproductive Fitness, and Fluctuating Asymmetry in Brook Stickleback: Responses to Anthropogenic Runoff

**Neal D. Mundahl** [1,*] and **Kelsey A. Hoffmann** [1,2]

1   Large River Studies Center and Southeastern Minnesota Water Resources Center, Departments of Biology and Geoscience, Winona State University, Winona, MN 55987, USA; kelsey.a.hoffmann@usace.army.mil
2   US Army Corps of Engineers, Saint Paul District, Rock Island, IL 61201, USA
*   Correspondence: nmundahl@winona.edu

**Abstract:** Multiple indicators have been used to assess the degree of exposure of fish to anthropogenic chemicals in their stream habitats. We hypothesized that brook stickleback (*Culaea inconstans*) in a headwater stream receiving urban and agricultural runoff (South Fork Whitewater River, SFWR) would exhibit poorer condition, reduced reproductive fitness, and a greater left side to right side morphological asymmetry (i.e., fluctuating asymmetry or FA) than fish from a nearby headwater stream with a forested drainage basin (Garvin Brook). Male and female fish were collected from both streams just prior to spawning in 2013–2015. In 2013 and 2014, fish were assessed for overall condition (Fulton's K), internal measures of condition (hepatosomatic index, HSI) and reproductive fitness (gonadosomatic index [GSI], total oocyte count, and oocyte mass). In 2015, measurements of head length, jaw length, eye diameter, pectoral fin length, and pelvic fin length were made on both sides of each fish for assessing degree of FA. We observed declining condition with fish size, increased liver size, and reduced oocyte counts and oocyte size in female brook stickleback in SFWR relative to those from Garvin Brook. SFWR females had significantly higher FA than Garvin females for all structures assessed, except pelvic fin length. FA also was slightly higher for all structures in SFWR males compared to Garvin males, but differences were not significant. A composite FA index combining all measurements from an individual fish into a single value displayed highly significant differences for female fish (SFWR FA >> Garvin FA), but not for male fish (SFWR FA = Garvin FA). Exposure of brook stickleback to reduced water quality in SFWR during early development appears to increase morphological asymmetry in female (but not male) fish, and continuing exposure to compromised water quality throughout life impacts both general condition and reproductive fitness of stickleback, especially older female fish, in SFWR.

**Keywords:** *Culaea inconstans*; fitness; condition; developmental asymmetry; reproduction

**Key Contribution:** Female brook stickleback display increased morphological asymmetry and reduced reproductive fitness after exposure to reduced water quality.

## 1. Introduction

Poor water quality can have significant effects on the health of aquatic organisms. Severe toxic spills, leaks, and releases of many substances can quickly kill entire communities of organisms in lakes, streams, and rivers [1–3], whereas milder conditions or events may cause more subtle, less obvious, but longer-term changes in organism behavior or fitness [4–7]. A wide range of human activities (e.g., logging, agriculture, mining, manufacturing, construction, and general urbanization) within watersheds have been shown to have negative impacts on many different types of aquatic organisms [4,7–9].

Like many regions worldwide, numerous lakes, streams, and rivers in Minnesota, USA, are affected negatively by human activities occurring within their respective drainage basins. These activities have resulted in widespread impacts on surface waters, with

Minnesota currently listing 2904 water bodies as impaired (with 6168 total impairments) due to compromised water quality [10]. Impairments encompass a wide range of water quality standards, including suspended sediments that reduce water clarity, excessive nutrients that increase algal production, bacteria that make water unsafe for contact or consumption, chemicals (e.g., mercury and PFOS) that can contaminate fish tissue and be passed on to humans when consumed, and generally unhealthy conditions for fish and aquatic macroinvertebrates [10].

Impaired water quality in streams and rivers may not directly kill aquatic fauna or prevent them from inhabiting certain systems, but their overall health and fitness may be significantly reduced [5,7,9,11]. Poor health and fitness in stream fish may be evident externally as small size caused by slower growth, lower mass at length, increased prevalence of DELT (deformities, eroded fins, lesions, and tumors) anomalies, or morphological asymmetry (i.e., fluctuating asymmetry, and FA) due to altered early development [5,11,12]. Internally, poor health and fitness in fish may be expressed as enlarged livers and smaller reproductive organs [5,12]. Such indicators of stress often are useful in detecting the presence of many pollutants at non-lethal, but chronic, levels [4,5,7].

In this study, we examined fish in two headwater streams in southeastern Minnesota to assess how differing degrees of water quality impairment may impact their fitness and reproductive condition. Specifically, we measured internal and/or external characteristics in brook stickleback (*Culaea inconstans*) just prior to their spawning season in three different years (expecting fish to exhibit peak fitness during these periods) to see if such variables responded to varying levels of stream impairment, and if they may be used to assess reduced water quality in other systems. Brook stickleback are broadly distributed in well-vegetated, coolwater stream margins and lake littoral zones across northern United States and southern Canada, where they feed on a variety of aquatic insects and micro-crustaceans [13]. Maturing in 1 year, these small (<70 mm TL) fish are highly territorial, building nests of various organic materials to protect their fertilized eggs and recently hatched larvae [13]. Although tolerant of both turbid waters and salinities >20 parts per thousand [13], the species has been shown to be physiologically and morphologically responsive to various chemical pollutants [11,14,15]. We expected that stickleback from a stream with more confirmed impairments resulting from agriculture and urbanization within the watershed would exhibit reduced condition and reproductive fitness and greater morphological asymmetry than fish collected from a nearby, but less impaired, stream.

## 2. Study Sites

Brook stickleback were examined in two headwater streams located 35 km apart in adjacent, separate watersheds in southeastern Minnesota, USA. The South Fork Whitewater River (SFWR) site was located within the City of Eyota, a small (population size = 2047) rural community surrounded by agricultural lands and livestock rangelands (Figure 1). The stream originates as two spring seeps that create wetlands bordered by agriculture fields 2 km upstream from the fish collection site within the city. Agricultural fields are drained by subsurface drain lines that connect to the stream, as well as by grassed waterways that conduct surface runoff toward the stream during heavy precipitation events or spring snowmelt. Within the city, the SFWR receives runoff via storm sewers from city streets, parking lots, and other impervious surfaces. In addition to brook stickleback, which comprised 90% of the fish community at this site during our study, creek chub (*Semotilus atromaculatus*) and western blacknose dace (*Rhinichthys obtusus*) were also present. During other years, central stoneroller (*Campstoma anomalum*) and southern redbelly dace (*Chrosomus erythrogaster*) have also been collected from the SFWR study site.

The Garvin Brook site was located within public forest lands approximately 4 km downstream from the headwater (limestone) springs (Figure 1). Agricultural fields, range-lands, and several rural residences also are located in the watershed upstream from our collection site, with all agriculture and rangelands located on bluff tops elevated 100 m or more higher than the headwater springs. A forested riparian buffer 100 to 200 m wide

borders both sides of the stream at the spring source and along the intermittent channel that extends for approximately 1 km above the headwater springs. The stream corridor between the spring source and our study site included public and private forest lands and a county park (comprised of forested lands and mowed turfgrass). Along with brook stickleback, brown trout (*Salmo trutta*), slimy sculpin (*Cottus cognatus*), and brook trout (*Salvelinus fontinalis*) were present at the Garvin Brook location.

**South Fork Whitewater River site**          **Garvin Brook site**

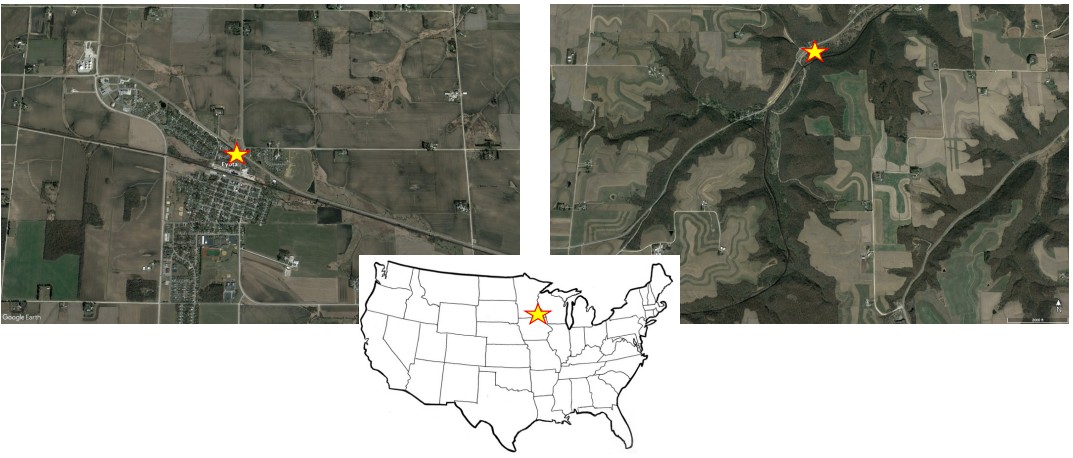

**Figure 1.** Aerial views of study site locations (indicated by yellow stars) on the South Fork Whitewater River (SFWR, 43°59′27.19″ N 92°13′46.68″ W) and Garvin Brook (44°00′29.64″ N 91°48′24.49″ W). The inset highlights the location of the study area (indicated by the yellow star) within the southeastern corner of Minnesota, USA. The SFWR site was located within the City of Eyota, a small community surrounded by agricultural land. The Garvin Brook site was located within state forest lands.

We relied on routine water quality monitoring data (physical, chemical, and biological) collected by the Minnesota Pollution Control Agency (https://www.pca.state.mn.us/air-water-land-climate/water-quality-trends-and-data, accessed on 10 November 2023) to document environmental conditions present at our two stream study sites. During the study period, the stream reach containing the SFWR site had four active water quality impairments (i.e., not meeting regional water quality standards) in effect: high fecal coliform bacteria count (>200 colony-forming units/100 mL), high turbidity (>10 NTU), reduced benthic macroinvertebrate community composition/structure (ranked as fair on a scale ranging from excellent to very poor), and reduced fish community composition/structure (fair). By comparison, the stream reach containing the Garvin Brook study site had only a single active water quality impairment: high fecal coliform bacteria count (>200 colony-forming units/100 mL). In addition, the SFWR immediately downstream from our study site was found to be contaminated with 13 different pharmaceutical and personal care products (PPCPs) during sampling in 2014, including three (bisphenol A, DEET, and caffeine) at concentrations sufficient to elicit biological responses (i.e., gene activation associated with a known toxicity responses in fish) [16]. SFWR water samples also exhibited estrogenic activity, with gene expression tests conducted on fish in SFWR water having impacts on genes associated with reproduction, development, growth, and tumor formation [16]. No screening of PCPPs in Garvin Brook has been conducted, although two nearby trout streams tested positive only for bisphenol A [16]. Several recent studies conducted on these two streams [3,17–22] further suggest that the SFWR has much poorer water quality than Garvin Brook.

## 3. Methods

Brook stickleback were collected from each study site during late May and early June, 2013–2015. These time periods were immediately prior to the onset of stickleback spawning in these waters. Fish were collected via a combination of backpack electrofishing and

overnight sets of standard cylindrical minnow traps "baited" with activated chemical light sticks. Sticklebacks were over-anesthetized with a 100 mg/L solution of tricaine methanesulfonate (MS-222), fixed for 48 h in an 8% formalin solution, and preserved and stored in 70% ethanol until examined.

Sticklebacks collected in 2013 and 2014 were used to assess fish condition and reproductive fitness. Fish were removed from the ethanol preservative, rinsed in running tap water for 1 to 2 min, then towel dried prior to weighing (wet mass, nearest 0.001 g) and measured (total length [TL], nearest mm). Masses and TLs were used to calculate a Fulton's condition factor (K, where $K = [\text{mass}/TL^3] \times 100{,}000$) for each individual fish [23]. Condition factors were compared between sites separately for each sex.

To assess reproductive fitness [24] and exposure to stressful environmental conditions [25], individual sticklebacks were dissected to remove gonads (ovaries in females and testes in males) and the liver. These organs were weighed (nearest 0.1 mg) to calculate a gonadosomatic index (GSI, where $GSI = [\text{mg gonad}/\text{mg total fish mass}] \times 100$) and a hepatosomatic index (HSI, where $HSI = [\text{mg liver}/\text{mg total fish mass}] \times 100$) for each fish. For female fish, oocyte number was estimated by doubling the oocyte count from a single ovary. Individual oocyte size (mass) was estimated by dividing combined ovary masses by total oocyte counts. After combining data from 2013 and 2014, ovary masses, testes masses, GSIs, HSIs, total oocyte counts, and oocyte sizes were compared between stream sites. Due to expected increases in gonad and liver masses, oocyte counts, and oocyte sizes with increasing fish size, we used analysis of covariance (ANCOVA) to control for fish size differences, to compare each of the fish length-internal fitness variable relationships between stream sites.

Stickleback collected in 2015 were anesthetized, preserved, rinsed, and towel dried as described above. Fish TL (nearest mm) was measured, and sex was determined based on external sex characteristics (e.g., dark coloration of males, lighter coloration and swollen abdomens [due to ovaries with developed oocytes] of females). FA was assessed for individual fish by taking duplicate measurements (nearest 0.1 mm) of five anatomical structures (lower jaw length, eye diameter, head length, pectoral fin length, and pelvic fin length; Figure 2) on both sides of the fish. Duplicate measurements were averaged prior to analysis. Differences in structure size between sides were calculated (left value minus right value) and compared to a value of 0 to test for "ideal FA" (i.e., mean values for each test group compared to 0 using t-tests, with separate tests for each structure/sex/stream). To compensate for variation in ideal FA among different structures (i.e., some displayed ideal FA, but others did not), FA for each structure was standardized (difference between left and right values/left value, expressed as %) and compared separately between sites for each sex. We also created a composite FA index for each fish, based on an average of the standardized values for all five structures assessed.

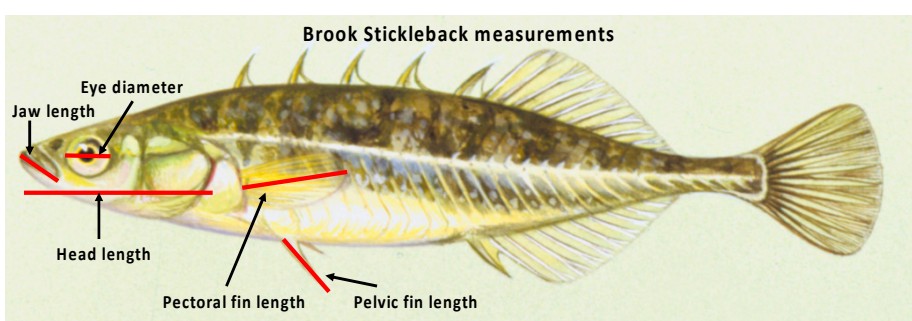

**Figure 2.** Side view of a brook stickleback depicting the five morphological measurements (jaw length, head length, eye diameter, pectoral fin length, and pelvic fin length) made on both left and right sides of each fish to assess fluctuating asymmetry.

## 4. Results

Across the three years, we examined 263 brook stickleback: 126 from Garvin Brook and 137 from SFWR (Table 1). Fish ranged in size from 26 to 69 mm TL, likely representing age 1 and age 2 fish [13]. All fish in both streams were sexually mature, based on examination of gonads (i.e., ovaries with large oocytes and enlarged testes). For all years combined, lengths of male and female fish did not differ within a stream, but both male and female fish were significantly larger in Garvin Brook than they were in SFWR (Figure 3A).

**Table 1.** Numbers (N) and total lengths (TL, mm) of male (M) and female (F) brook stickleback collected from Garvin Brook and the South Fork Whitewater River (SFWR) for assessments of general and reproductive fitness, 2013–2015.

| Stream | Sex | 2013 | | 2014 | | 2015 | | Total N |
| --- | --- | --- | --- | --- | --- | --- | --- | --- |
| | | **N** | **TL range** | **N** | **TL range** | **N** | **TL range** | |
| Garvin | M | 31 | 28–66 | 15 | 31–49 | 20 | 35–69 | 66 |
| | F | 16 | 29–65 | 15 | 34–52 | 29 | 26–61 | 60 |
| SFWR | M | 26 | 34–60 | 15 | 35–59 | 34 | 29–61 | 75 |
| | F | 26 | 34–62 | 15 | 35–64 | 28 | 34–62 | 62 |
| Total N | | 92 | | 60 | | 111 | | 263 |

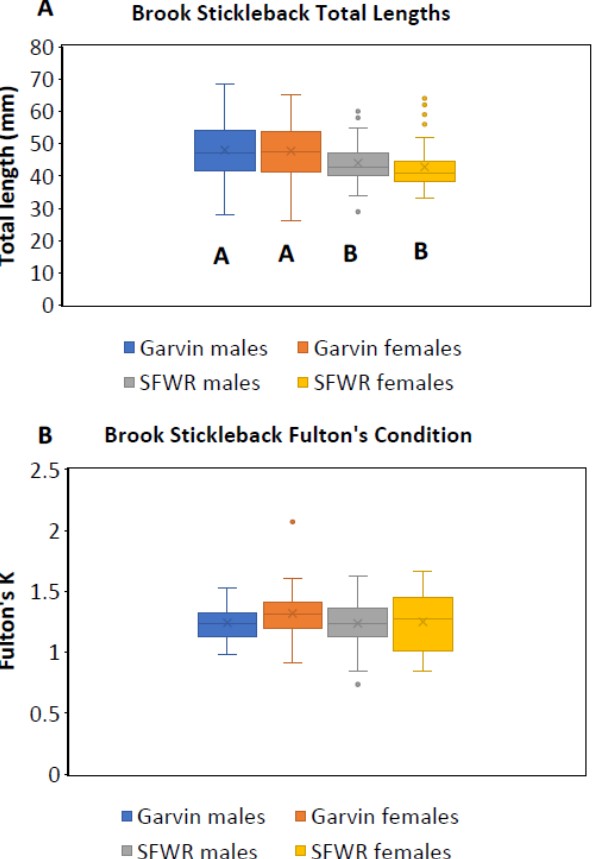

**Figure 3.** Box and whisker plots (means, medians, interquartile ranges, standard deviations, and outliers) of total lengths ((**A**), 2013–2015) and Fulton's condition factors ((**B**), 2013–2014) for male and female brook stickleback from Garvin Brook and South Fork Whitewater River (SFWR). Total length bars not sharing a common letter beneath them are significantly different (ANOVA and least significant difference tests).

Overall condition of stickleback, based on Fulton's K, did not differ between streams or between sexes when data from 2013 and 2014 were examined (Figure 3B), averaging 1.26 across streams and sexes. Although there were no significant changes in Fulton's K with increasing fish length in either male or female fish from Garvin Brook (Figure 4A,B), female stickleback in SFWR displayed significantly declining condition with increasing fish length and male fish displayed a similar, nearly statistically significant decline (Figure 4C,D).

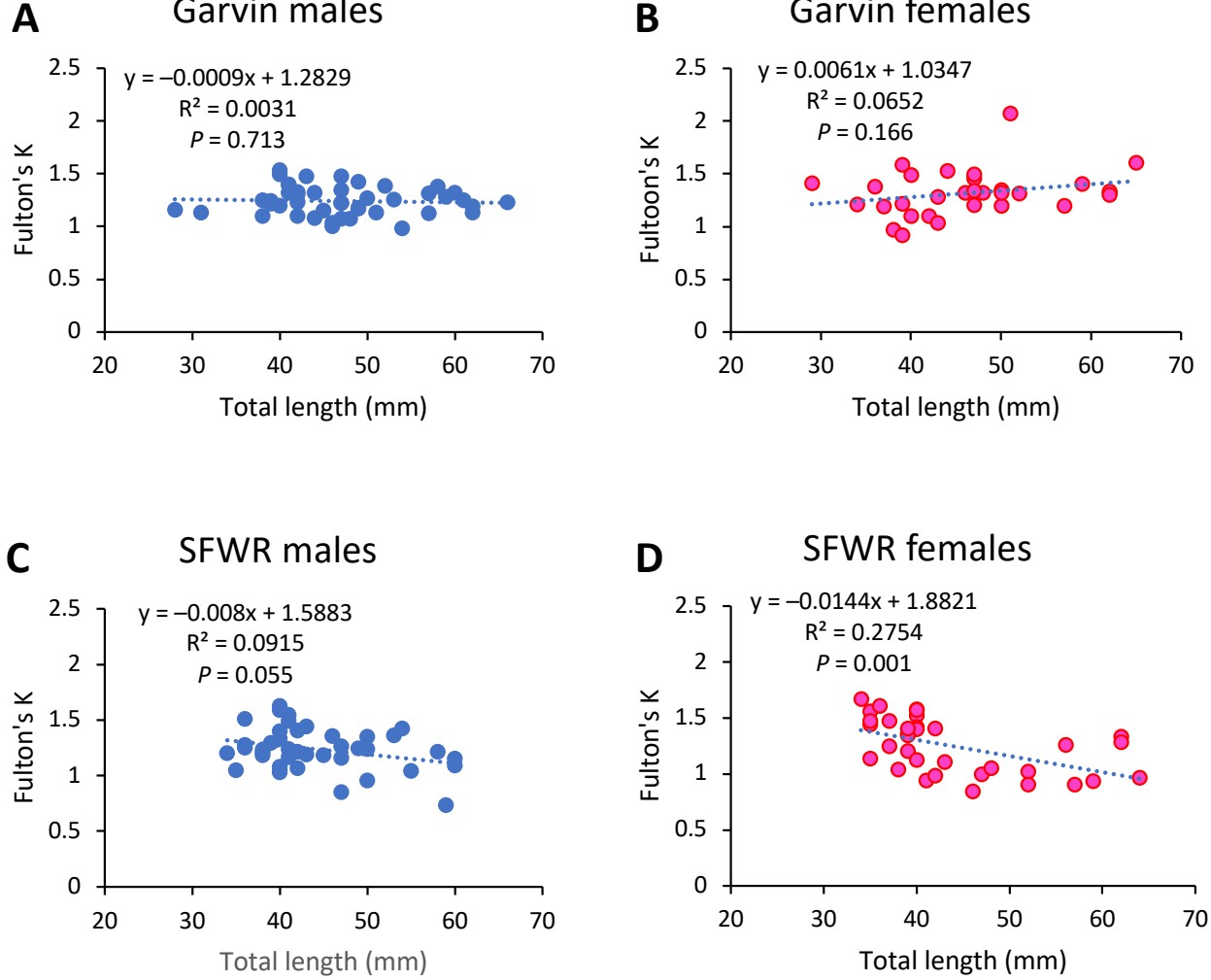

**Figure 4.** Relationships between male (**A,C**) and female (**B,D**) brook stickleback total lengths and Fulton's condition factors (K) from Garvin Brook and South Fork Whitewater River (SFWR), 2013–2014. Simple linear regression statistics are displayed for each stream/sex combination.

ANCOVA tests comparing internal measures of reproductive fitness and condition in stickleback (ovary and testes masses, GSI, HSI, oocyte number, and mass) between the two study streams revealed several differences during the years examined (Table 2). Neither testes weights (Figure 5A) nor male GSI values (Figure 6A) differed significantly (Table 2) between the two streams after accounting for fish size. However, ovary masses (Figure 5B) and female GSI values (Figure 6B) displayed variable patterns in the different streams. Slopes of the total length–ovary mass regressions significantly differed between sites (Table 2), invalidating the ANCOVA test but demonstrating that larger females in Garvin Brook had larger ovaries than females of similar lengths in SFWR (Figure 5B). In contrast, ANCOVA revealed that Garvin Brook females had significantly greater GSI values than those from SFWR (Table 2, Figure 6B). ANCOVA also revealed significantly higher oocyte numbers in Garvin versus SFWR females across the entire range of fish size (Table 2, Figure 7A). Slopes of the total length–oocyte mass regressions differed significantly between

sites (Table 2), again invalidating the ANCOVA test but demonstrating that larger females in Garvin Brook had larger oocyte than females of similar lengths in SFWR (Figure 7B). Finally, ANCOVA for both male and female HSI (Table 2) indicated significantly higher HSI values (i.e., enlarged livers that represented a greater proportion of total body mass) in SWFR fish compared to Garvin fish (Figure 8A,B).

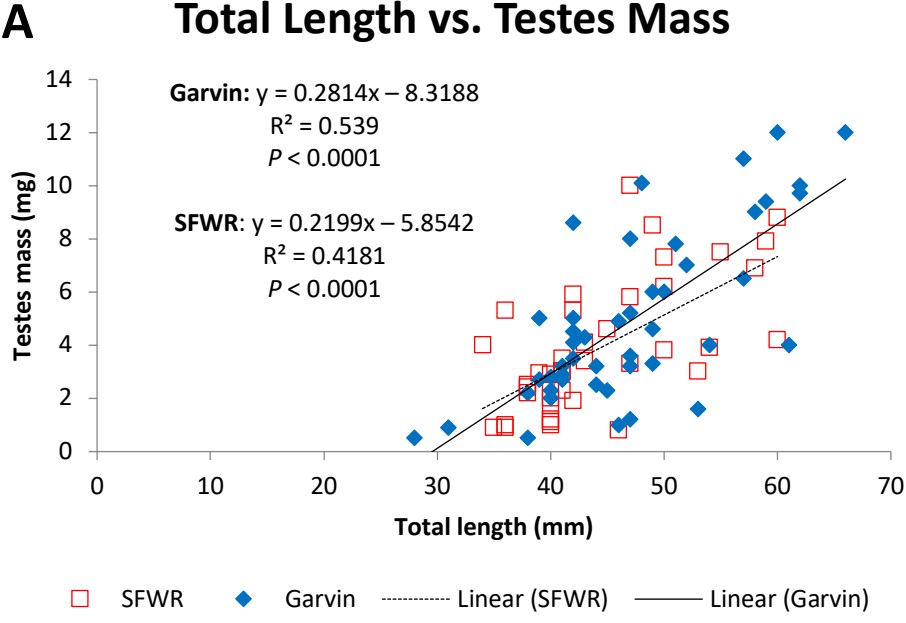

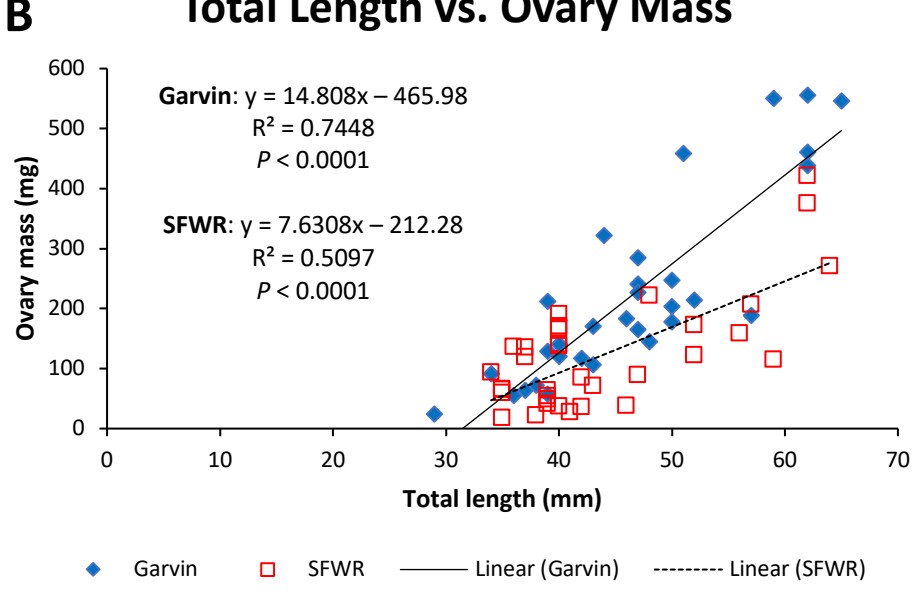

**Figure 5.** Total length–gonad mass relationships for male (**A**) and female (**B**) brook stickleback from Garvin Brook and South Fork Whitewater River (SFWR), 2013–2014. Simple linear regression statistics are displayed for each stream/sex combination. Refer to Table 2 for results of ANCOVA comparisons between streams.

**Table 2.** Analysis of covariance (ANCOVA) and homogeneity of slopes test results comparing internal reproductive fitness and general condition variables in brook stickleback between Garvin Brook and South Fork Whitewater River sites, 2013 and 2014. Fish total length was used as a covariate. Significant ($p < 0.05$) tests are highlighted in **bold font**. GSI = gonadosomatic index, HSI = hepatosomatic index.

| | ANCOVA Test | | Homogeneity of Slopes | |
| --- | --- | --- | --- | --- |
| **Variable** | ***F* Value** | ***p* Value** | ***F* Value** | ***p* Value** |
| Testes masses | 0.54 | 0.464 | 1.11 | 0.295 |
| Ovary masses | **12.63** | **<0.001** | **12.02** | **0.001** |
| Male GSI | 0.04 | 0.842 | 0.01 | 0.921 |
| Female GSI | **10.39** | **0.002** | 0.14 | 0.710 |
| Oocyte numbers | **4.02** | **0.049** | 0.00 | 1.000 |
| Oocyte masses | **11.85** | **0.001** | **5.87** | **0.018** |
| Male HSI | **23.82** | **<0.001** | 2.35 | 0.129 |
| Female HSI | **9.87** | **0.003** | 1.46 | 0.232 |

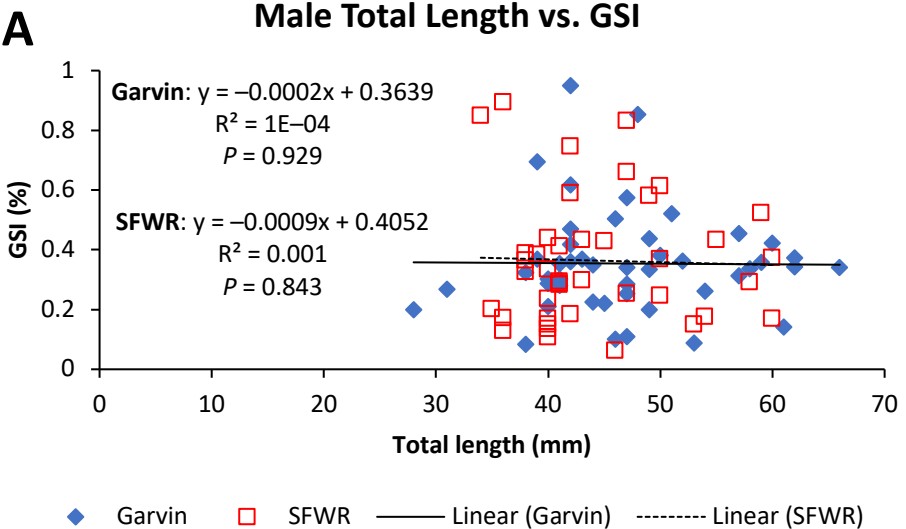

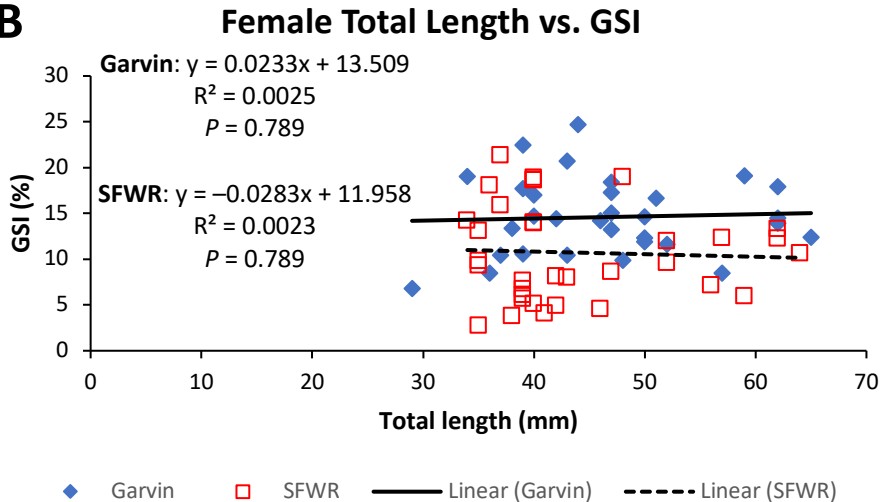

**Figure 6.** Total length–gonadosomatic index (GSI) relationships for male (**A**) and female (**B**) brook stickleback from Garvin Brook and South Fork Whitewater River (SFWR), 2013–2014. Simple linear regression statistics are displayed for each stream/sex combination. Refer to Table 2 for results of ANCOVA comparisons between streams.

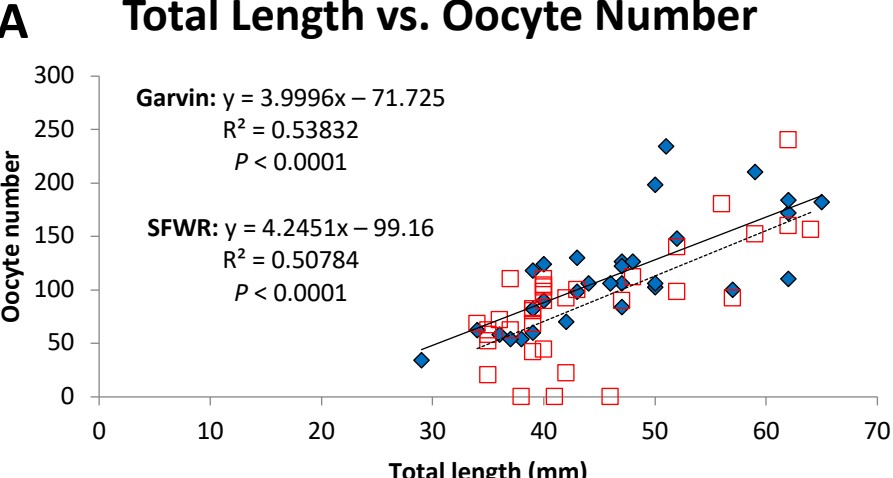

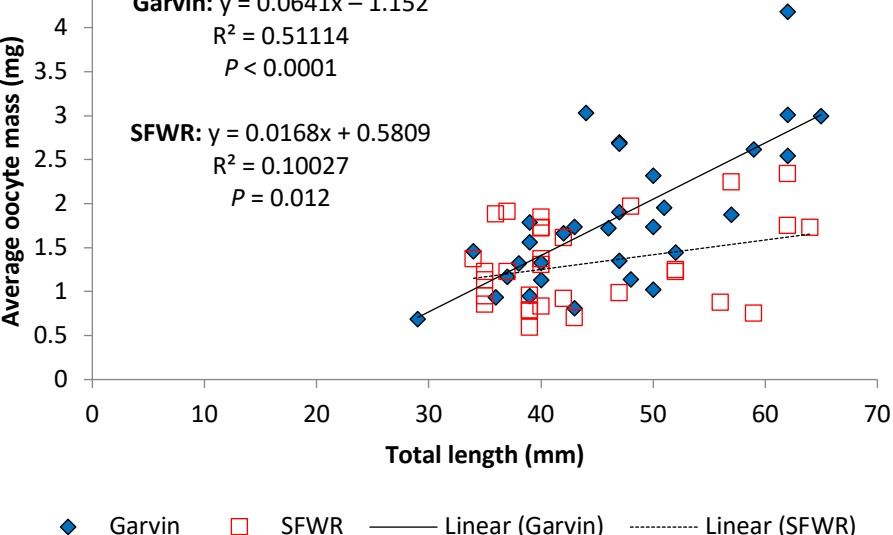

**Figure 7.** Total length–oocyte number (**A**) and total length–oocyte mass (**B**) relationships for female brook stickleback from Garvin Brook and South Fork Whitewater River (SFWR), 2013–2014. Simple linear regression statistics are displayed for each stream. Refer to Table 2 for results of ANCOVA comparisons between streams.

FA in Garvin Brook and SFWR stickleback, as assessed by left side–right side differences in head, jaw, eye, pectoral fin, and pelvic fin measurements, exhibited high variability during 2015 (Figure 9). Nine of the twenty stream/sex/structure categories examined did not display ideal FA, i.e., mean values significantly differed ($p < 0.05$) from 0 (Figure 9). Seven of these nine categories skewed positive (left-side structures more often longer than right-side structures), and six were observed in male stickleback. Due to these variations in ideal-versus-not ideal FA among the structures examined, we standardized all side-to-side differences for each individual fish prior to assessing possible differences in FA between streams.

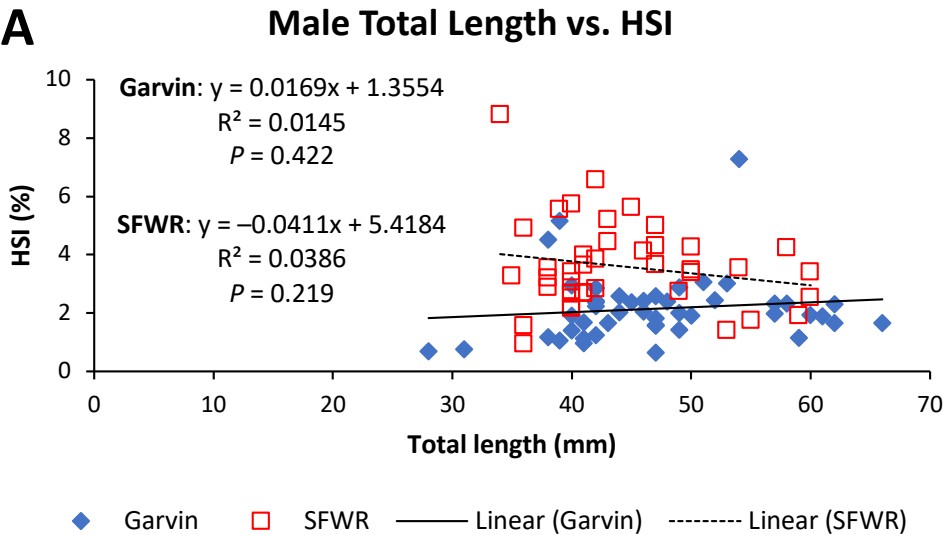

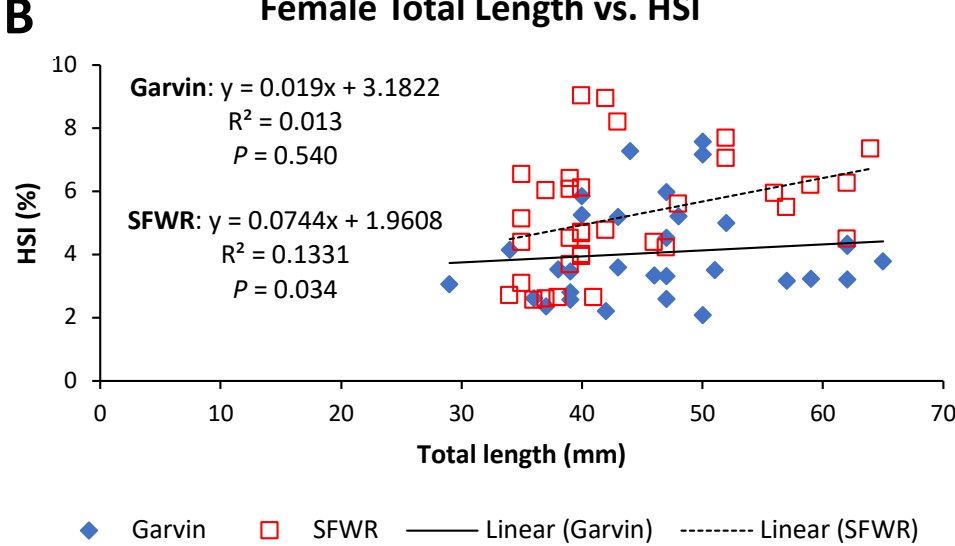

**Figure 8.** Total length–hepatosomatic index (HSI) relationships for male (**A**) and female (**B**) brook stickleback from Garvin Brook and South Fork Whitewater River (SFWR), 2013–2014. Simple linear regression statistics are displayed for each stream/sex combination. Refer to Table 2 for results of ANCOVA comparisons between streams.

Mean FA measures for all five structures examined in this study were higher for both male and female brook stickleback from SFWR than from Garvin Brook (Figure 10). These differences were significant for four of the five female characters (jaw length, head length, eye diameter, and pectoral fin length) plus a composite index (Figure 10A). However, none of the male characters nor the male composite index displayed significant differences between Garvin Brook and SFWR (Figure 10B). Consequently, SFWR female stickleback displayed significantly higher degrees of FA than female fish from Garvin Brook, whereas male fish from both systems exhibited similar degrees of FA.

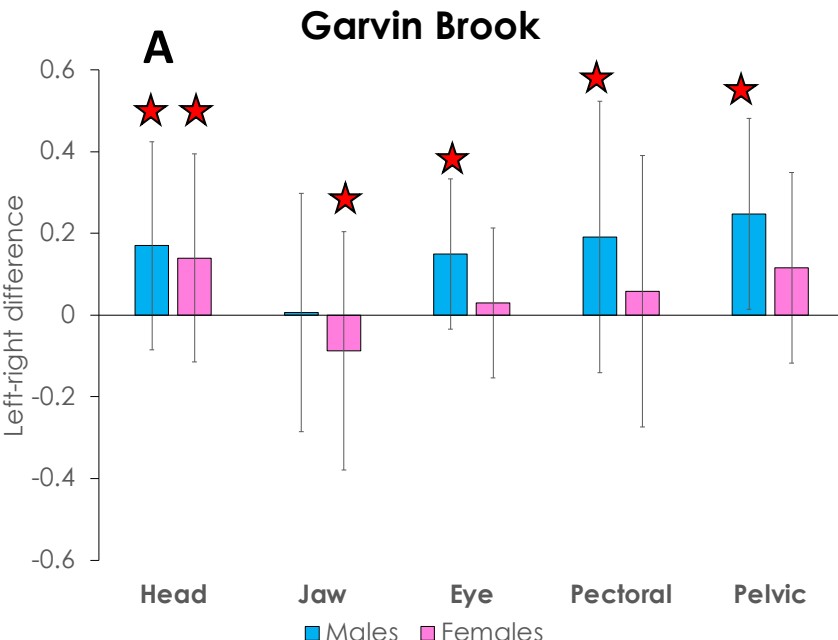

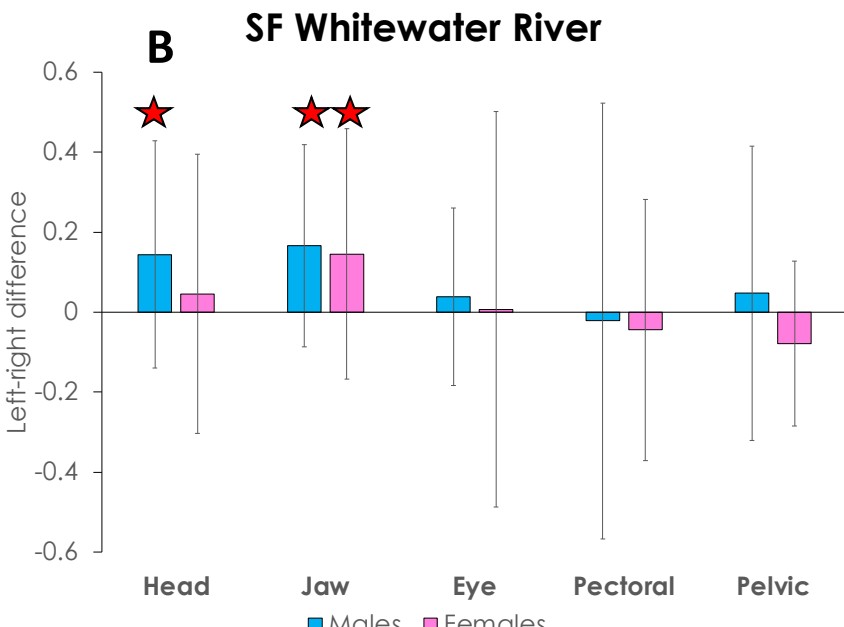

**Figure 9.** Left side versus right side differences in measurements of five morphological structures (head length, jaw length, eye diameter, pectoral fin length, and pelvic fin length) in male and female brook stickleback from Garvin Brook (**A**) and South Fork Whitewater River (**B**), 2015. Bars are means, and vertical lines represent ±1 standard deviation. Means highlighted by stars are skewed significantly different from 0 (i.e., do not display ideal fluctuating asymmetry).

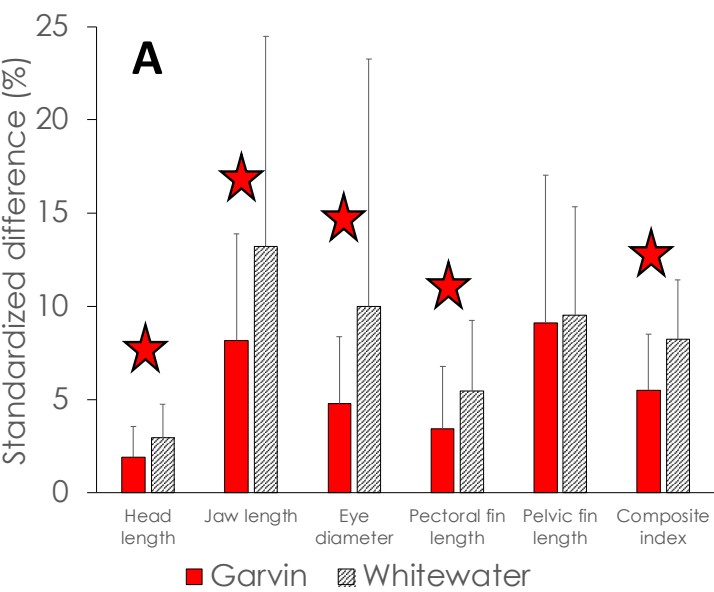

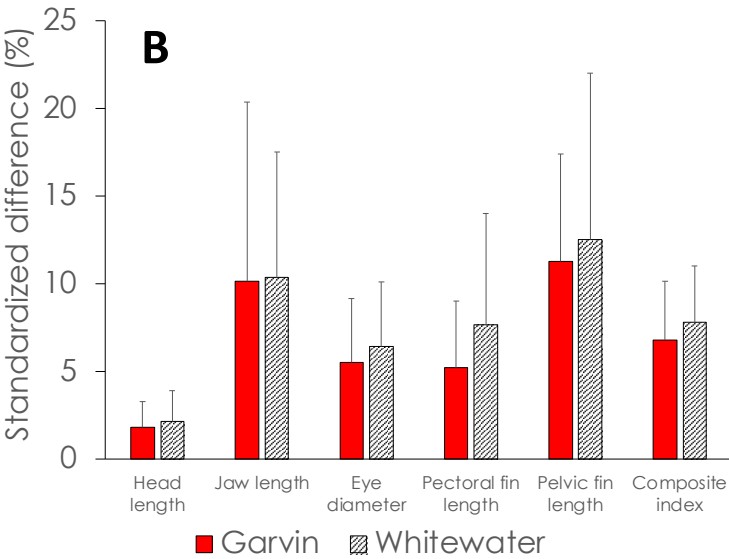

**Figure 10.** Standardized fluctuating asymmetry (difference between left side versus right side measurements, expressed as % of left side measurement) for five morphological structures (head length, jaw length, eye diameter, pectoral fin length, and pelvic fin length) plus a composite index (see description in Methods) in female (**A**) and male (**B**) brook stickleback from Garvin Brook and South Fork Whitewater River, 2015. Bars are means and vertical lines represent ±1 standard deviation. Between-stream comparisons highlighted by stars are significantly different.

## 5. Discussion

Our study of brook stickleback condition and reproductive fitness in two headwater streams in Minnesota provides evidence that various indicators of stickleback health likely can be used to demonstrate fish exposure to impaired water quality. We found that stickleback in the more impaired SFWR displayed smaller size, declining Fulton's condition with increasing fish size or age, and enlarged livers compared to stickleback from the less impaired Garvin Brook site. Female fish from SFWR also displayed smaller ovaries and smaller oocytes, plus greater FA than females in Garvin Brook.

Brook stickleback morphology appears to respond to impaired water quality in several predictable ways. Previous studies [11,12,14,15] have reported that various stickleback species are sensitive, both physiologically and morphologically, to environmental perturbations. Typically, a relatively small number of freshwater aquatic species (e.g., fathead minnow *Pimephales promelas*, rainbow trout *Oncorhynchus mykiss*, cladocerans such as *Daphnia magna*, midges *Chironomus tentans*, scuds *Hyallela azteca*, and zebra fish *Danio rerio*) are favored as models or bioindicators during traditional laboratory and/or field investigations used to assess the potential effects of various chemicals on aquatic life [26]. But other species, including brook stickleback, have been presented as either more sensitive bioindicators or more widely distributed and consequently more applicable over a broader geographic area [11,15,27] than many of the typical test species, especially under field conditions. Consequently, the results of our study add support to the recommendation [11] for using brook stickleback as a model for detecting environmental impacts on freshwater aquatic life.

Smaller size, reduced condition in larger (older) fish, and enlarged livers all suggest that brook stickleback in the SFWR were exposed to reduced water quality that produced some degree of chronic stress. Fish exposed to many pollutants often respond similarly to what we observed in stickleback in the SFWR [4,5,7,14]. Pollution stress can slow or suppress growth, leading to smaller overall body sizes for fish living in polluted waters [28–32]. Long-term exposure to poor water quality also may lead to reduced fish condition (e.g., lower Fulton's K) in older fish, as low-level but continuing stress may reduce foraging ability/success, especially for older fish with higher maintenance needs [22,33,34]. The same impairments stressing fish may also suppress prey resources (i.e., fewer potential prey) [19] due to poor water quality, making it more difficult for stickleback to find and consume sufficient prey to remain healthy.

Both male and female sticklebacks in the SFWR had significantly higher HSI values compared to fish in Garvin Brook. Enlarged livers (hepatomegaly) are often indicative of contaminant stress in fishes [11,25,35–37]. Exposure to waterborne pollutants can induce greatly increased metabolic activity within the liver as it works to detoxify contaminants taken in through the gills or food, producing larger liver mass relative to body mass [25]. Since livers of female fish also enlarge during egg development due to the organ's role in vitellogenesis [38], male fish (which do not undergo vitellogenesis except under rare conditions) provide better models for using relative liver size to assess exposure to water pollutants [25,39]. In our study, SFWR males averaged 67% higher HSI than Garvin males, whereas SFWR females averaged 30% higher values than Garvin females. These data suggest that fish in the SFWR had been likely exposed to more impaired water quality than those in Garvin Brook, and even vitellogenesis occurring in female fish did not mask the difference in liver size between the two systems.

The smaller oocytes and ovaries present in female stickleback in the SFWR point toward some form of stress experienced by these fish during oogenesis. Oocyte development and growth in fishes is sensitive to many different environmental conditions, ranging from water temperature and oxygen levels to the presence of many different chemical pollutants [4,5,32]. Female fish are especially susceptible to estrogen-like chemicals that may either stimulate or suppress various reproductive processes [4,5,11,32] or interfere with sex determination [28]. The reduced GSI values for female stickleback in the SFWR are parallel to the declines observed in female stickleback GSIs after exposure to estradiol in the laboratory [11]. However, in some fish species the testes of males may be more impacted by some pollutants than the ovaries of the females, primarily due to the pollutants' ability to alter various spermatogenesis processes [5]. It could be argued that smaller female fish in general produce fewer and smaller oocytes than do larger females [40,41], complicating interpretation of the effects of water quality on stickleback oocyte number and size. However, our results demonstrated that female stickleback in the more impaired waters of the SFWR produced significantly fewer and smaller oocytes than females of the same size in Garvin Brook, especially when the largest females were examined. Consequently,

various reproductive characteristics of female brook stickleback appear to be sensitive to water quality, allowing for their use as bioindicators of pollutant exposure.

Elevated FA in female (but not male) brook stickleback in the SFWR demonstrates that some factor(s) impacted the early development of female embryos, interfering with the normal processes responsible for producing perfect bilateral symmetry. Many studies have reported that fish and amphibians exposed to various waterborne chemicals develop increased asymmetry, demonstrating the sensitivity of embryonic development to environmental stressors [7,9,14,15,42]. Several of the same anatomical structures measured in our study have been used when assessing FA in other fish species, and these structures have been shown to be impacted by poor water quality [7,9,14]. Increased FA in fish and other organisms is generally considered as a response to environmental stress, caused when a developing organism is forced to allocate more energy to stress response and has less energy to expend on developmental control [6,15].

FA has been criticized by some researchers as an inconsistent approach for assessing the presence of environmental stressors [6,15]. Criticisms include lack of repeatability, susceptibility of the procedure to sampling and measurement error, alternative causes of asymmetry not related to organismal stress, sample size issues, and others. FA methodologies and data analyses have been modified over time to compensate for several of these concerns [6,9,15,43]. Our study of brook stickleback used several of these modified approaches, and we were successful at demonstrating a consistent pattern of greater asymmetry in most of the characters assessed in female fish in SFWR (but not in male fish) relative to those in Garvin Brook.

Female fish in general may be more sensitive to environmental stressors, especially those water-quality impairments that are chemically similar to estrogen [4,5,11,28,32]. There also appear to be correlations between asymmetry and both oocyte number and size in female stickleback, with fish that exhibit asymmetry producing fewer and smaller oocytes than those produced by symmetric individuals [14]. These correlations are likely not cause-and-effect, but instead are separate, organismal (physiological and developmental) responses to a common environmental stressor. In our study, we observed both reduced oocyte number/size and greater FA in female fish from the more impaired SFWR, although these were not assessed simultaneously in the same individual fish. However, these observations in female stickleback, coupled with limited to no responses (i.e., lack of FA and no change in testes weights or GSI values) in male fish to the poorer water quality in the SFWR, support the contention that females are the more sensitive gender and have greater potential use as a bioindicator.

## 6. Conclusions

Our study of brook stickleback in two streams differing in water quality indicated that exposure of brook stickleback to reduced water quality from agricultural and urban runoff during early development resulted in increased morphological asymmetry in female (but not male) fish. In addition, continued exposure to compromised water quality throughout life impacted both general condition and reproductive fitness of stickleback, especially older female fish. These results corroborate previous studies [11,14,15] which suggested that brook stickleback can serve as a bioindicator of water quality impairment, and this indicates that the species can be used as an effective environmental monitor within its natural habitats across its wide geographic range in North America [11,15].

**Author Contributions:** N.D.M. and K.A.H. developed the concept for the paper. N.D.M. carried out all field collections, and N.D.M. and K.A.H. conducted the laboratory measurements. N.D.M. and K.A.H. analyzed the data and N.D.M. led the writing of the paper. All authors have read and agreed to the published version of the manuscript.

**Funding:** Partial funding for this project was provided by the Trout Unlimited Driftless Area Restoration Effort.

**Institutional Review Board Statement:** Fish collections were carried out under special permits (Numbers 18903, 19721, 20320) from the Minnesota Department of Natural Resources, Division of Fish and Wildlife, Section of Fisheries, and were conducted with the approval of the Winona State University Animal Care and Use Committee (1317064-1, 1310072-2). This research complied with all ethical standards.

**Data Availability Statement:** Data available from lead author upon reasonable request.

**Acknowledgments:** We thank the City of Eyota and the Minnesota Department of Natural Resources, Division of Forestry, for granting us access to the stream sites. We thank Jeff Hastings, Trout Unlimited, for supporting research on non-game animals in trout streams.

**Conflicts of Interest:** The authors declare no competing interests.

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
