# Peer review of "Condition, Reproductive Fitness, and Fluctuating Asymmetry in Brook Stickleback: Responses to Anthropogenic Runoff"

_fishes, doi:10.3390/fishes8110557_

Round 1
Reviewer 1 Report
Comments and Suggestions for Authors
The authors have done a study to quantify the effects of water pollution on various quantifiable fish attributes, such as weight, different lengths, and gonadal measures.
The findings are statistically significant and will enhance our comprehension of the effects of water pollution on fish. Moreover, it offers us a fish that may be utilized to ascertain the quantity of pollution in a water body.
The introduction is well-crafted, providing a comprehensive background and presenting a logical and attainable premise.
The map depicted in Figure 1 has the potential to be significantly larger in size. I recommend that the authors provide a single comprehensive map that clearly indicates both streams.
I strongly suggest that the authors provide the readers with a table encompassing various water parameters and pollutant parameter data for both streams, even if it necessitates the utilization of supplementary data with proper referencing. This would enable the readers to comprehend the results with more clarity.
I suggest that the authors utilize n-values for each figure.
Figure 5 has captivated my interest. It would have been intriguing if the authors had included a critical threshold, possibly by regression analysis, to determine the scale at which the effects of water pollution become more noticeable.
I am particularly curious to determine if there is any correlation between the size at sexual maturity of the species and the observed variances. It would be intriguing to ascertain the histological alterations in gonadal development as well.
How can the authors elucidate the notable disparities in FA measurements between SFWR and Gravin specifically for females, but not observed in males?
Do mature /gravid females typically have lower Fulton's K values because they allocate more energy towards reproduction rather than growth during the reproductive season?
Reviewer 2 Report
Comments and Suggestions for Authors
Reviewer 3 Report
Comments and Suggestions for Authors
The work has potential for publication, but I suggest major revisions in the methodology and presentation of the figures. All suggestions are in the attached PDF.

Minor editing of English language required
Round 2
Reviewer 3 Report
Comments and Suggestions for Authors
The present manuscript may be considered for publication in this journal. I currently have no other comments about the study. I agree with the publication.